# Lilac (*Syringa oblata*) genome provides insights into its evolution and molecular mechanism of petal color change

Bo Ma [1,2,6], Jing Wu[1,6], Tian-Le Shi[2,6], Yun-Yao Yang[1], Wen-Bo Wang[1,2], Yi Zheng[1], Shu-Chai Su[2], Yun-Cong Yao[1], Wen-Bo Xue[3], Ilga Porth[4], Yousry A. El-Kassaby [5], Ping-Sheng Leng [1✉], Zeng-Hui Hu [1✉] & Jian-Feng Mao [2✉]

Color change during flower opening is common; however, little is understood on the biochemical and molecular basis related. Lilac (*Syringa oblata*), a well-known woody ornamental plant with obvious petal color changes, is an ideal model. Here, we presented chromosome-scale genome assembly for lilac, resolved the flavonoids metabolism, and identified key genes and potential regulatory networks related to petal color change. The genome assembly is 1.05 Gb anchored onto 23 chromosomes, with a BUSCO score of 96.6%. Whole-genome duplication (WGD) event shared within Oleaceae was revealed. Metabolome quantification identified delphinidin-3-O-rutinoside (Dp3Ru) and cyanidin-3-O-rutinoside (Cy3Ru) as the major pigments; gene co-expression networks indicated *WRKY* an essential regulation factor at the early flowering stage, *ERF* more important in the color transition period (from violet to light nearly white), while the MBW complex participated in the entire process. Our results provide a foundation for functional study and molecular breeding in lilac.

[1] Beijing Advanced Innovation Center for Tree Breeding by Molecular Design, College of Landscape Architecture, Beijing Laboratory of Urban and Rural Ecological Environment, Bioinformatics Center, Beijing University of Agriculture, Beijing 102206, China. [2] Beijing Advanced Innovation Center for Tree Breeding by Molecular Design, The Key Laboratory of Silviculture and Conservation of the Ministry of Education, National Engineering Research Center of Tree Breeding and Ecological Restoration, Key Laboratory of Genetics and Breeding in Forest Trees and Ornamental Plants of Ministry of Education, College of Forestry, College of Biological Sciences and Technology, Beijing Forestry University, Beijing 100083, China. [3] BGI Genomics, BGI-Shenzhen, Shenzhen 518083, China. [4] Département des Sciences du Bois et de la Forêt, Faculté de Foresterie, de Géographie et Géomatique, Université Laval, Québec, QC G1V 0A6, Canada. [5] Department of Forest and Conservation Sciences, Faculty of Forestry, University of British Columbia, Vancouver, BC V6T 1Z4, Canada. [6] These authors contributed equally: Bo Ma, Jing Wu, Tian-Le Shi. ✉email: lengpsh@tom.com; buahuzenghui@163.com; jianfeng.mao@bjfu.edu.cn

As one of the most important ornamental traits, floral color diversity has not only attracted much attention in the breeding and development of new varieties, but also play an important role in the evolution of floral traits and ecological effects[1]. Flower color change is a common phenomenon that accompanies the developmental period. The flower color of many plants is formed early and then fades during flower opening. For example, *Malus hupehensis*[2] and *Brunfelsia acuminata*[3] have red and purple flowers respectively in early flower opening stage, which gradually change to white with the opening. In the petals of *Chrysanthemum morifolium*[4] and *Cleome hassleriana*[5], the heavy pink coloration and anthocyanin content show the highest level before flower opening, and then the color gradually changed to light pink due to the continuous decrease in anthocyanin content. In addition, with the flower development the petal color of *Paeonia* 'Coral Sunset' and 'Pink Hawaiian Coral'[6] fade from coral to light pink to pale yellow, presenting color group change. Petal color changes is often accompanied by both accumulation and loss of anthocyanins; therefore, a better understanding of the metabolic dynamics and molecular basis of petal color change is important for prolonging flower coloring and improving ornamental and economic value.

The accumulation of floral color has been studied more intensively and is mainly determined largely by distinct pigments, including flavonoids, carotenoids and/or alkaloids[7]. Among which, flavonoids as the third-largest group of natural products, are widely known for their important roles in plant fitness, health benefits and food quality[8]. Anthocyanin biosynthesis, as part of the flavonoids pathway, is essential for the appearance of these colors, which are expressed in plant tissues in a wide variety, from red to purple and even blue[9]. As we know, pigment accumulation is mainly determined by biosynthetic genes and their transcriptional regulation via transcription factors (TFs), which plays a major role in flower color development[10,11]. Currently, advances in genetic engineering make rare flower color development and directed breeding possible[11,12].

Actually, there have been many studies on the biosynthesis and regulatory mechanisms of flower color, yet the widespread phenomenon of petal color change has been neglected. The genus *Syringa* (Oleaceae), known for the outstanding beautiful corollas, consists of more than 27 species and 2000 cultivars worldwide, of which 22 are distributed in China, including 18 endemic species[13]. As the representative species, *Syringa oblata* (lilac) is native to China with more than 1000 years of cultivation[13], owing to its elegant colors, unique fragrance, beautiful corollas, large inflorescence, and early flowering. Moreover, as one of the rare trees with purple flowers, *S. oblata* is the most widely cultivated as ecological and ornamental tree, spanning subtropical, warm temperate, temperate, and the edge of the cold temperate zones[14]. Since the 20th century, there have been more than 200 new cultivars bred with *S. oblata* as excellent hybrid parents, including *Syringa* × *hyacinthiflora* and cultivars breeding by crossing between *S. oblata* and *S. vulgaris*. It is observed that lilac petals are darkest and red-purple in the bud stage, then gradually change to violet and finally fade close to white. Differences in anthocyanins lead to flower color changes and the production of various flower colors[11]; however, the biochemical and molecular basis for this change is still unclear. Lilac is considered as an excellent material to study the mechanism of petal color change because of its phenotypic advantages.

In lilac flowers, only two anthocyanins have been reported in the full blooming stage[15], so a comprehensive assay and the metabolic dynamic analysis is lacking. Several studies have delved into the expression difference of some synthesis genes in the flavonoid biosynthetic pathway during flower developmental stage and cloned and verified the functions of some genes[16], however, with

**Table 1 Major indicators of the *S. oblata* genome.**

| Parameter | Number/Size |
|---|---|
| Estimated genome size (Gb) | 1.16 |
| Assembled genome size (Gb) | 1.12 |
| Chromosome-anchored scaffolds (Gb) | 1.05 |
| Karyotype (chromosomes, 2n) | 46 |
| N50 of contigs (bp) | 3,956,172 |
| Number of contigs | 443 |
| Longest contigs (bp) | 12,690,304 |
| N50 of scaffolds (bp) | 46,330,164 |
| Longest scaffolds (bp) | 69,209,139 |
| GC content (%) | 34.5 |
| Complete BUSCOs (%) | 96.6 |
| Repeat content (%) | 54.3 |
| Number of predicted genes | 35,313 |
| Number of annotated genes | 33,268 (94.21%) |
| Gene length_average/median (bp) | 4515.34/3336 |
| Number of miRNA | 231 |
| Number of tRNA | 724 |
| Number of rRNA | 138 |
| Number of snRNA | 3690 |

the absence of genomic evidence, it is difficult to explore their regulatory mechanisms[16]. Therefore, the whole-genome sequence of *S. oblata* is needed to develop an in-depth understanding of the evolution and potential regulatory networks of petal color formation underlying biochemical process in successive flower development stages[12].

Here, we present a high-quality chromosome-scale genome assembly of *S. oblata*, and it was determined by a combination of long-read sequencing and Hi-C scaffolding technologies. In total, 1.05 Gb and 93.75% of the assembled genome was anchored on 23 chromosomes with a scaffold N50 of 46.33 Mb. Furthermore, we predicted to contain 35,313 protein-coding genes and clarified the species WGD history. The metabolic dynamic results systematically revealed the composition and relative quantification of compounds related to flavonoid biosynthesis in lilac petals. Moreover, combining gene annotation, transcriptome and metabolome evaluations, we revealed a time-ordered co-expression network of color-changing petals, and identified the enzyme genes and potential regulatory factors controlling the key pigments associated with petal color change. This reference genome sequence, metabolic dynamic results and the generated gene regulatory network for petal color change are expected for further functional genomics investigations and molecular breeding.

## Results

**Genome sequencing, assembly, and quality assessment.** Genome sequencing resulted in a total of 237.19 Gb of nanopore long-reads (Supplementary Fig. 1; Supplementary Table 1), with approximately 204-fold depth of 1.16 Gb genome, as estimated based on *K*-mer frequency analysis (Supplementary Fig. 2; Supplementary Table 2). Furthermore, short-reads PCR-free sequencing generated 45.65 Gb (39-fold) reads (Supplementary Table 3); from which 154.31 Gb (133-fold) Hi-C paired-end reads were produced (Supplementary Fig. 3; Supplementary Table 3). A high genome heterozygosity of 1.92% and a moderate repeat rate of 51.79% were assessed by 25-mer analysis (Supplementary Fig. 2). Here, we obtained an assembly ready for Hi-C scaffolding, which was 1.12 Gb in the length of contig with an N50 length of 3.96 Mb (Supplementary Table 4).

With the Hi-C scaffolding, the final assembly was generated by anchoring 443 contigs to 23 chromosome-scale super scaffolds with a total length of 1.05 Gb (Table 1; Supplementary Table 5; Supplementary Fig. 3), accounting for 93.75% (1.05/1.12Gb) of

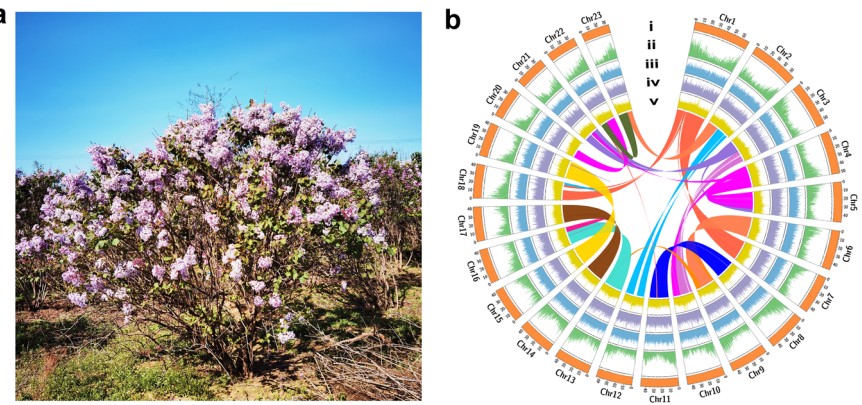

**Fig. 1 Image and genome characterization of *Syringa oblata*. a** Photo of *S. oblata* in Beijing, China (by Bo Ma, 2021). **b** Twenty-three pseudochromosomes were ordered by size. Chr1-Chr23 refers to the 23 assembled chromosomes. (i) Assembled chromosomes (Mb, orange); (ii) Gene density (green); (iii) Transposon density (blue); (iv) Tandem repeat density (purple); (v) GC content (yellow); Line in the center of the circle represent chromosome collinearity.

the assembly, with a scaffold N50 of 46.33 Mb (Supplementary Table 5). The number of pseudochromosomes matched the experimentally determined number of chromosome[17] (2n = 46) (Fig. 1) and the pseudomolecules ranged in size from 34.29 to 69.21 Mb (Supplementary Table 6).

The short-reads genome mapping rate was 99.37%. Further, 1559 complete BUSCO (96.60%) were assembled, among which only 27 (1.70%) had fragmented matches, and 28 (1.70%) were entirely missing (Supplementary Table 7), indicating that the assembled genome had high accuracy, contiguity and completeness.

**Genome annotation**. A combination of three different strategies was used to perform the repeat annotation, including ab initio, homology-based, and transcriptome-based predictions. In total, we identified 572,494,098 bp repeat elements, accounting for 54.30% of the assembled genome (Table 1; Supplementary Table 8). This proportion is higher than that reported for other sequenced plants of Oleaceae, including *Fraxinus excelsior*[18] 35.90%, *Osmanthus fragrans*[19] 49.35%, and *Olea europaea*[20] 42.95%. These repeat elements contained 43.91% predominantly known transposable elements (TEs) (38.87% retrotransposons and 5.04% DNA transposons), 6.33% unclassified TEs, 3.71% tandem repeat, and 0.34% low complexity. Among all TEs, 37.79% consisted of LTR-RTs and most of them are LTR/Gypsy, occupying 23.37% of the genome, followed by the LTR/Copia (12.54%) (Supplementary Table 8).

A total of 35,313 protein-coding genes were predicted (Table 1), with average gene length of 4515 bp, of which 33,268 (94.21%) were assigned functional annotation (Supplementary Table 9), and 91.77% of the predicted genes had complete structure (Supplementary Table 10). The average length of transcripts and coding regions were 1650 bp and 1148 bp, respectively, with a mean of 6.41 exons (Supplementary Table 9). In total, 30,454 (86.24%), 13,431 (38.03%), 15,733 (44.55%), 33,242 (94.14%), 25,882 (73.29%), 29,765 (84.29%), and 29,833 (84.48%) gene models were annotated to Cluster of Orthologous Groups of proteins (COG)[21], Gene Ontology (GO)[22], Kyoto Encyclopedia of Genes and Genomes (KEGG)[23], the NCBI Non-Redundant Protein Sequence Database (NR), Swiss-Prot, TAIR, and MSU Rice Genome Annotation Project Database[24], respectively (Supplementary Table 9). Among which, in the GO annotation, 6681 genes were assigned to the biological process, 10,653 genes were assigned to the cellular compound, and 6958 genes were assigned to the molecular function categories. Additionally, we identified non-coding RNAs, including 231 miRNAs, 724 transfer

RNAs (tRNA), 138 ribosomal RNAs, and 3690 small nuclear RNAs (snRNA) (Table 1; Supplementary Table 11).

We identified 3035 genes encoding TFs from 95 TF families in the lilac genome, accounting for 8.59% of protein-coding genes. The most members in the TF families were WD40 (308), AP2/ ERF-ERF (205), MYB (180), bHLH (159), C2H2 (144), NAC (142), and WRKY (117).

**Gene family analysis and phylogenetic tree construction**. To investigate the evolution of the lilac genome, we performed a comparative genomic analysis with three Oleaceae plants (*O. europaea*, *F. excelsior*, and *O. fragrans*), three sequenced flowering plants (*Prunus mume*, *Rhododendron delavayi*, and *Carica papaya*), two representative plants of Solanaceae family (*Solanum lycopersicum* and *S. tuberosum*), two rosid plants (*Arabidopsis thaliana* and *Vitis vinifera*), and a monocot plant (*Oryza sativa*) (Supplementary Table 12). A total of 29,856 gene families encompassing 386,920 genes were identified among the 12 species, from which 14,829 gene families (33,058 genes) were found in *S. oblata* (Supplementary Table 13). The sequences of the 356 conserved single-copy orthologous genes identified were used to reconstruct the phylogenetic relationship with *O. sativa* serving as the outgroup (Fig. 2a). The revealed phylogeny was in agreement with previous studies[20,25], which indicated that *S. oblata* had the closest relationship with Oleaceae (*F. excelsior*, *O. fragrans*, and *O. europaea*) and formed a clade with Solanaceae (*S. lycopersicum* and *S. tuberosum*) followed by *R. delavayi*, and was distant from the other species. In addition, lilac was separated from the Oleaceae common ancestor at ~ 43.20 (38.23–48.18) million years ago (Mya) (Fig. 2a).

We also identified unique and shared gene families among these species. A total of 11,952 gene families were shared with Oleaceae, and 2982 unique gene families were found in *S. oblata* genome compared with *O. europaea* (5590), *F. excelsior* (3174), and *O. fragrans* (3625) (Fig. 2b); 400 gene families (2255 genes) were found to be specific to the assembled *S. oblata* genome when compared with the other 11 genomes (Supplementary Table 13). Moreover, 990 gene families (3876 genes) were found in expansion, while 1599 gene families (3513 genes) were found in contraction in the *S. oblata* lineage (Fig. 2b). Functional enrichment analysis showed that expanded gene families were enriched in GO terms mostly involved in 'binding', 'transport', 'metabolic process', and 'signal transduction' (Supplementary Fig. 4). In contrast, KEGG enrichment analysis revealed that the expanded genes were mainly from pathway implicated in plant

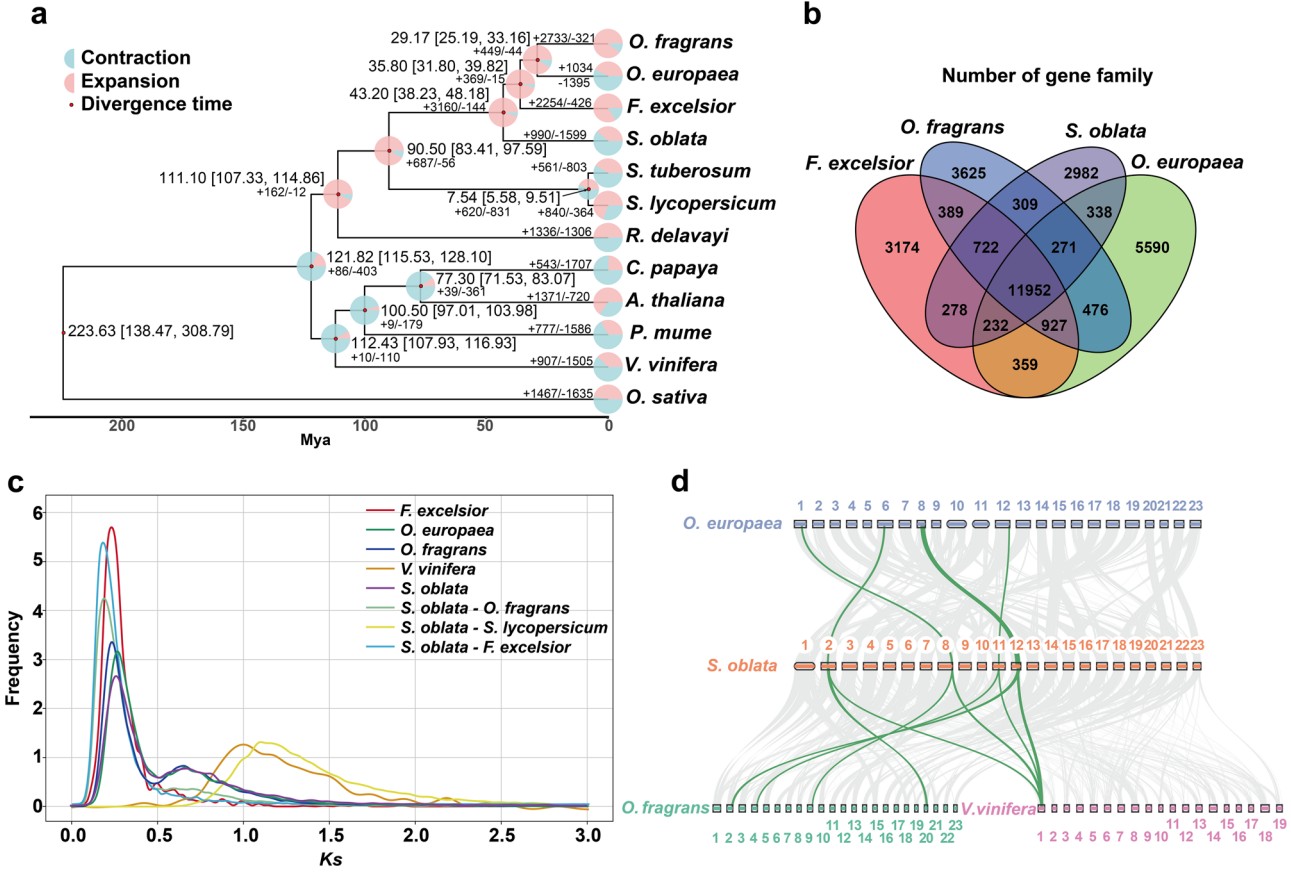

**Fig. 2 Lilac genome evolution. a** Phylogenetic tree, gene family expansions/contractions in lilac and 11 other species based on a concatenated alignment of 356 single-copy orthologs, and divergence times. **b** Venn diagram illustrating the shared and unique gene families among *S. oblata* and three Oleaceae plants (*O. europaea, F. excelsior,* and *O. fragrans*). **c** *Ks* distribution of *S. oblata, F. excelsior, O. europaea, O. fragrans, S. oblata-F. excelsior, S. oblata-O. fragrans, S. oblata- S. lycopersicum* and *V. vinifera*. **d** Chromosome collinearity analysis among lilac, *O. europaea, O. fragrans* and *V. vinifera*. Green lines represent four syntenic blocks in lilac match up to one region in *V. vinifera*.

defense response, development, and monoterpenoid biosynthesis (Supplementary Fig. 5).

**Gene synteny analysis and WGD events.** Synonymous substitutions per site (*Ks*) age distribution, synteny analyses, and inferred dating times were used to unveil WGD events and to date the duplication events[26]. The divergence time of lilac and the other three Oleaceae species was approximately predicted at 43.20 Mya (*Ks* = 0.2) (Fig. 2a), simultaneously, they produced the same signature *Ks* peaks of WGDs at about 0.25[19,20,27] (Fig. 2c), higher than 0.2, suggesting that the most recent WGD event, estimated at around 54 MYA (*Ks* = 0.25), occurred before the divergence of Oleaceae family (Fig. 2c, Supplementary Fig. 4). Similarly, the main *Ks* peak value of 0.25, was smaller than the *Ks* peak suggesting divergence between Oleaceae and Solanaceae (approximately 1.1) (Fig. 2c), indicating that the most recent WGD event occurred after the divergence of Oleaceae and Solanaceae. Moreover, the intra-genomic synteny analyses showed that 15,786 gene pairs distributed along 855 blocks in lilac (Supplementary Fig. 6; Supplementary Table 14), with synteny analyses of *O. fragrans* (Supplementary Fig. 7; Supplementary Table 14), which further supported this Oleaceae-specific WGD event. At the same time, the inter-genomic analyses between *S. oblata* and *O. fragrans* exhibited relatively conservative collinearity (Fig. 2d; Supplementary Fig. 8; Supplementary Table 14), supporting a close evolutionary relationship.

We also observed a weaker signal of an additional ancient WGD shared by Oleaceae lineage (Fig. 2c), which has been previously

reported based on genomic data[20]. Subsequently, we detected a 4:1 syntenic relationship between *S. oblata* and *V. vinifera* (Fig. 2d), and built a gene homology dot plot (Supplementary Fig. 9; Supplementary Table 14), which provided additional evidence for the identified two WGD events[19,20] after the Gamma paleohexaploidy in the stem lineage of core eudicots[28,29]. Moreover, the *Ks* analysis of four Oleaceae plants supported the WGD events (Fig. 2c), which were inferred from the lilac genome (as well as from *O. europaea*[20], *F. excelsior*[18] and *O. fragrans*[19]) and were specific to Oleaceae.

**Reconstruction of flower pigments biosynthesis pathway in *S. oblata*.** Our high-quality genome assembly and transcriptome data allowed reconstruction of the flower pigment biosynthesis pathway, specifically capturing the implicated enzymatic genes in this process. We unveiled 124 genes encoding functioning enzymes in the flavonol (from colorless to pale yellow) and colored anthocyanin (from pink-red to violet) biosynthesis pathway (Fig. 3); 14 important gene families in this pathway were identified, among which, 4CL (19), F3H (15), DFR (15), and F3oGT (24) gene families had a large number of members (Fig. 3). In addition, we found that members of the PAL, F3H, F3'H|F3'5'H, DFR, F3oGT and RT gene families were replicated, especially F3H and F3oGT, which might enhance flavonoids production. Gene expression profiles showed that during the flower opening period the *CHS* (*So16G045740* and *So19G016880*), *F3H* (*So13G050140* and *So15G003400*), and *F3'H|F3'5'H* (*So05G012500* and *So07G050530*) increased firstly and then

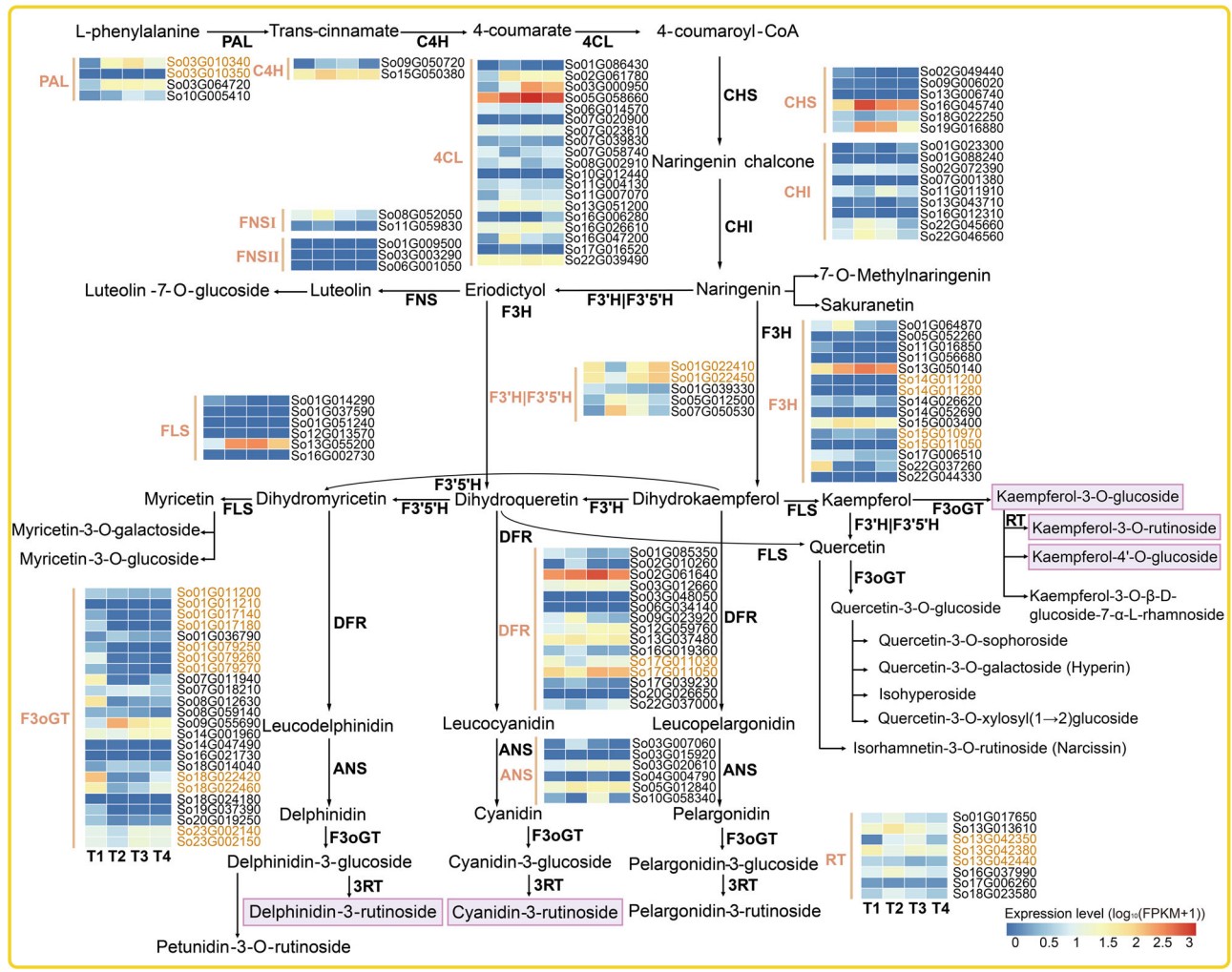

**Fig. 3 The metabolic pathways and metabolic composition of flavonoids in *S. oblata*.** The metabolic pathways of flavonol and anthocyanin. Gene expression profiles (log₁₀(FPKM+1)) at different flower opening stages (from left to right were T1-T4), and the bar color change from blue to red indicates the expression from low to high. The left side of the heat map was the corresponding enzymes, and the right was the genes id within the genome. Genes identified in tandem clusters were marked in yellow color, and the top5 metabolites of petals at the four periods were marked with purple box. PAL phenylalanine ammonia-lyase, C4H cinnamate-4-hydroxylase, 4CL 4-coumarate CoA ligase 4, CHS chalcone synthase, CHI chalcone isomerase, F3H flavanone 3-hydroxylase, F3'H flavonoid 3'-hydroxylase, F3'5'H flavonoid 3'5'-hydroxylase, FLS flavonol synthase, DFR dihydroflavonol 4-reductase, ANS anthocyanidin synthase, F3oGT flavonoide-3-O-glucosyltransferase, RT rhamnosyltransferase.

decreased, and *F3H* (*So17G006510* and *So22G037260*) and *F3'H|F3'5'H* (*So01G039330*) gradually decreased, which was consistent with the pattern of anthocyanin contents (Figs. 3 and 4b; Supplementary Fig. 11).

Delphinidin-based anthocyanins which had the most contents in lilac petals usually were the major constituents of violet and blue flowers in nature, and F3'5'H was the key synthesis enzyme[30]. We noticed that the *So07G050530* expression first increased and then decreased during the flower opening period, which showed the high levels at T2 and T3 (Fig. 3). Its expression profile was consistent with blue-violet anthocyanins petunidin-3-O-rutinoside (Pt3Ru) contents (Fig. 4b), suggesting that the expression of *So07G050530* may be related to Pt3Ru synthesis.

**Metabolic dynamic results revealed the petal color change in *S. oblata*.** Before blooming, there was a rapid pigment accumulation process in the lilac flower, and gradually decreased during the flower opening[16], with color change from dark to light purple (Fig. 4a). For further study the lilac's petal color change, four flower opening period were divided according to phenotypic

characteristics (Fig. 4a, T1-T4). At the T1 stage (flower bud stage), the flower buds were red-purple and close to the calyx, no corolla tubes and lobes were showing or even barely exposed; at T2 (initial flower opening stage), corolla tubes were elongated, flower buds were enlarged, corolla lobes were about ready to open, and flower color was from red-purple to violet; at T3 (full flower opening stage), corolla lobes were stretched with almost 90° angle between the corolla lobes and tubes, and flower color was light violet; and at T4 (last flower opening stage), flowers color decayed and wilted, the corolla lobes were almost reversely curled, and the color of the flower became lighter than other three stages and tend to be in light violet nearly white (Fig. 4a).

A total of 118 flavonoid metabolites (Supplementary data 1) were detected, including 9 anthocyanins and 40 flavonols, and anthocyanins accumulation was the main factor determining flower color. The top 10 compounds accounted for 71.86% of the total content, including 2 anthocyanins, Dp3Ru (tended to be violet-blue[31], 22.76%) and Cy3Ru (tended to be red-purple[31], 14.33%) (Fig. 4b); 4 kaempferol-based flavonoids, kaempferol-3-O-rutinoside (8.33%), kaempferol-4'-O-glucoside (5.11%), kaempferol-3-O-glucoside (4.83%), and kaempferol-3-O-β-D-glucoside-7-α-L-rhamnoside

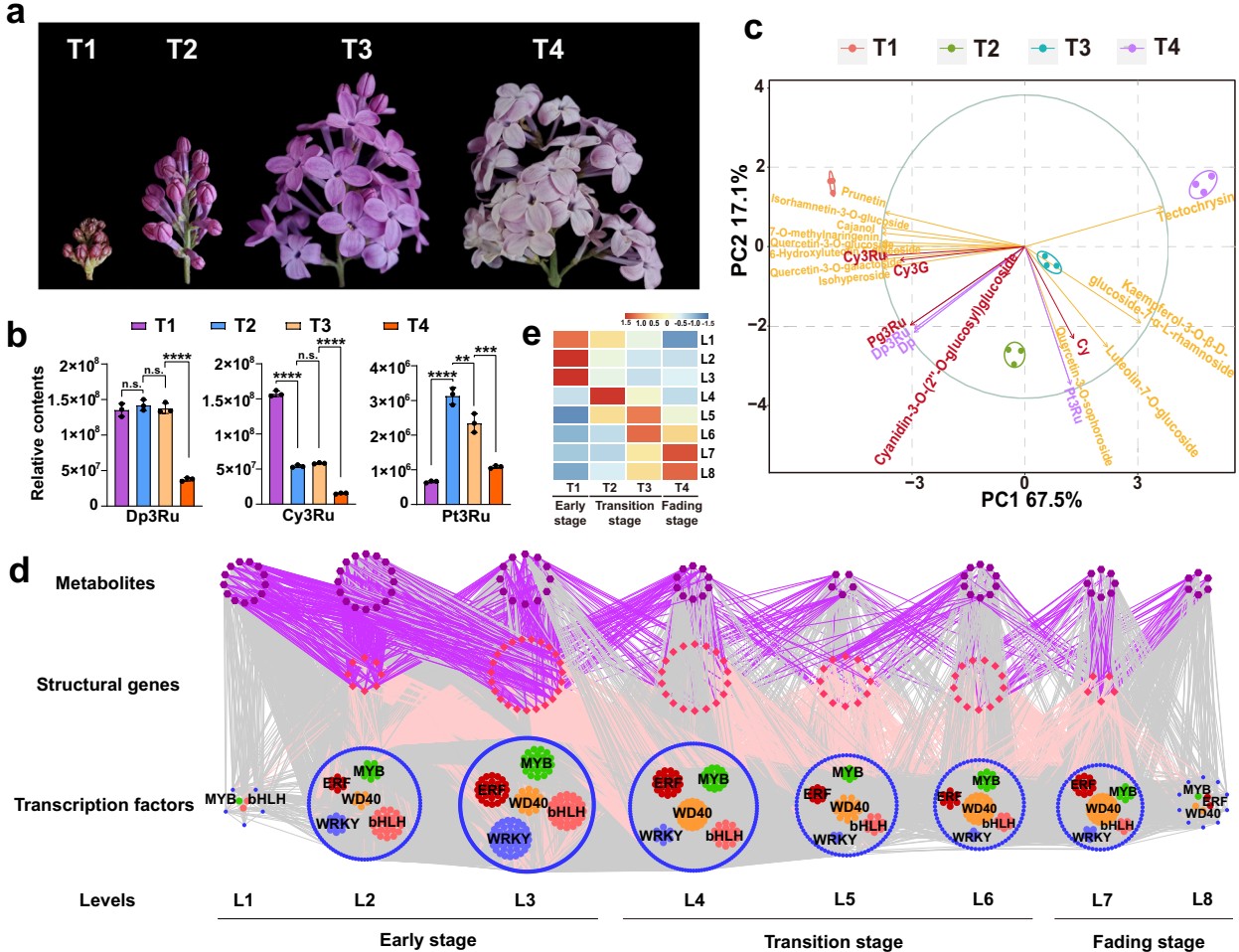

**Fig. 4 Principal component analysis and time-ordered gene co-expression network related to flower coloring. a** Four flower opening stages of lilac from T1 to T4 (photographed by Bo Ma in 2021). **b** Relative contents of main anthocyanins including Dp3Ru, Cy3Ru, and Pt3Ru at the four flower opening stages. Data are presented as mean ±SE, $n = 3$ biologically independent samples (Supplementary Data 2); statistical significance was calculated by one-way ANOVA with multiple comparisons between adjacent stages, and was indicated by asterisks, in which n.s. (no significance), $p < 0.05$ (*), $p < 0.01$ (**), $p < 0.001$ (***), and $p < 0.0001$ (****). **c** PCA distance biplot. The PCA biplot shows both the PC scores (PC1 and PC2 here) of samples (here: three petal samples for each of the four flower opening periods, as dots) and the loadings of variables (here: the relative content of metabolites in each petal sample, as vectors) in the 2-dimension space defined by the first two PCs. We can observe the difference between petal samples at different flower opening periods, the relative contribution of metabolite variables to this difference, and the relationship of the difference. **d** Predicted regulatory network and the connection among metabolites, structural genes of flavonoids (Fig. 3) and TFs. The purple dots on the top represent metabolites; the pink dots in the middle represent structural genes in the flavonoid metabolic pathway (Fig. 3); at the bottom, inside the blue circle, the green dots represent the *MYB* genes, the red dots represent the *ERF* genes, the orange dots represent the *WD40* genes, the pink dots represent the *bHLH* genes, the blue dots represent the *WRKY* genes; and outside of the circle, blue dots represent the remaining TFs. L1 to L8 indicate the levels identified in the time-ordered gene co-expression network. The PCC value increases from small to large, the width of the edges changes from thinner to thicker in the networks. **e** the heatmaps of average normalized FPKMs for each flower opening periods at each level identified in the time-ordered gene co-expression network. The three stages of flower color change are determined based on the gene expression profile and metabolome, as early stage (T1, red-purple), transition stages (T2-T3, violet), and fading stage (T4, light violet nearly white).

(4.49%); 4 quercetin-based flavonoids, isorhamnetin-3-O-rutinoside (3.66%), isohyperoside (3.03%), quercetin-3-O-xylosyl(1→2)glucoside (2.73%), and quercetin-3-O-galactoside (2.57%) (Supplementary Fig. 10; Supplementary data 1). For the top 5 metabolites (Fig. 3), except for kaempferol-3-O-rutinoside, the contents of remaining 4 metabolites yielded the lowest values at T4 (Supplementary Fig. 10), consistent with flower color changes as seen with phenotypic observation (Fig. 4a). The latter three flavonoids (kaempferol-3-O-rutinoside, kaempferol-4-O-glucoside, and kaempferol-3-O-glucoside) determined the pure yellow flower color, and showed an overall trend of first increasing and then decreasing, all of which reached the highest contents at T2 (Supplementary Fig. 10).

Subsequently, we focused on the 9 anthocyanins, namely delphinidin-based (4 types, tended to be violet-blue), cyanidin-based (4 types, tended to be red-purple), and pelargonidin-based (1 type, tended to be orange-red) anthocyanins (Fig. 4b; Supplementary Fig. 11). The first two, Dp3Ru and Cy3Ru determined lilac flower color and showed the highest contents in T1; and at T2 and T3, the contents of Dp3Ru was still high, while Cy3Ru only showed one-third of content at T1 (Fig. 4b). In addition, the content change of pelargonidin-3-O-rutinoside (Pg3Ru) was consistent with Dp3Ru, which was much higher than the others, and cyanidin-3-O-glucoside (Cy3G) was consistent with Cy3Ru (Supplementary Fig. 11). Different from the monotonously decreased contents of first three pigments,

the Pt3Ru, a delphinidin-based anthocyanin, first showed an increasing pattern and then decreased along with flower opening, with the highest value at T2 (Fig. 4b). Interestingly, the differential accumulation of pigments demonstrated that the cyanidin-based anthocyanins caused the color change of lilac from T1 to T3, while Dp3Ru did not differ significantly among the first three flower opening stages, causing color changes from T3 to T4 (Fig. 4b).

Furthermore, the PCA based on 20 differentially expressed major metabolites, including 8 anthocyanins (marker in red and purple in Fig. 4c; Supplementary Fig. 11) and 12 flavonoids with high contents (marker in yellow in Fig. 4c; Supplementary Fig. 12), showed that the first two principal components (PCs) cumulative contribution of the accounted for 84.6% of the variation (Fig. 4c). It is worth noting that PC1 (accounted for 67.5%) and can separate all four stages (Fig. 4c). Furthermore, the samples were arranged clearly in order of flower opening time (from left to right, Fig. 4c), with which anthocyanins, the higher contents at the early flower opening stage (Cy3Ru, Cy3G, Dp3Ru, Pg3Ru, and delphinidin (Dp)) were negatively correlated, while Pt3Ru and cyanidin (Cy) were positively correlated. As can be seen, the metabolic dynamics of these two pigments, Cy3Ru and Pt3Ru, were closely related to the color change of lilac flowers at the early and late blooming stages, respectively (Fig. 4b–c). Therefore, PCA clearly revealed the dynamic changes of the anthocyanin and other flavonoids contents with flower opening stages, which was an important basis for the dynamic integration of flower color phenotype and relevant gene expression.

**Dynamic regulation of petal color change in _S. oblata_.** The 12 samples of the four stages were clearly separated by metabolome and transcriptome PCA analysis (Supplementary Fig. 13). Between any two flower opening stages (T1 to T4), we found a total of 20,984 differentially expressed genes (DEGs) (2507 TFs and 18,477 structural genes) with an average Fragments Per Kilobase Million (FPKM) greater than 0.5 and 80 metabolites exhibiting significant differentiation. Then, a TF gene of MYB family, So16G008570, expressing in a high level at T1 and monotonically decreased until T4, was selected as the initial node to build the time-ordered gene co-expression network (TO-GCN). According to Pearson's correlation coefficient (PCC) values, eight time-ordered expression levels (denoted as L1 to L8, Fig. 4d) were obtained, which comprised 137,523 edges and 1427 nodes. Finally, 1347 genes including 1268 significantly differentially expressed TFs genes and 79 structural genes of anthocyanin and flavonol biosynthesis pathway, and 80 metabolites (including 8 anthocyanins) made up the TO-GCN (Fig. 4d). These 8 levels matched the expression peaks of DEGs at the four flower opening stages, as shown by the red squares (high expression levels) along the diagonal in the heat-map, which formed the basis for the inference of upstream and downstream genes/metabolites regulatory relationships (Fig. 4e).

The analysis revealed that most network nodes appeared at T1 (corresponding to time-ordered levels L1-L3), including 31 enzymatic genes in the anthocyanin and flavonol biosynthesis, 612 TFs, and 41 metabolites. There were 265 genes (including 247 TFs), 10 metabolites at T2 (corresponding to time-ordered level L4), and 287 genes (including 262 TFs), 15 metabolites at T3 (corresponding to time-ordered levels L5-L6). The least genes, only 5 structural genes appeared at T4 (corresponding to time-ordered levels L7-L8), when there were 147 TFs and 14 metabolites. The distribution of structural genes in time-ordered levels showed that the early biosynthetic genes (EBGs: PAL, C4H, 4CL, CHS, CHI, F3H, F3'H)[8] of flavonoid

biosynthesis pathway were mainly clustered in the latter levels (Fig. 4d).

For better description, we further distinguished the early flower opening stage (T1, when flowers were red-purple), the transition stage (T2 and T3, violet), and the fading stage (T4, nearly white). At T1, we found that among the 583 TFs that may directly regulate structural genes of the flavonoid pathway, the bHLH family contained the most members (59), followed by the WRKYs family (53) (Fig. 5a). In addition, in the potential regulatory sub-network (Fig. 5b) the structural genes were regulated mostly by MYB, bHLH, and WD40, including _F3oGT_ (So01G017180; So01G079250; So01G079260; So01G079270; So18G022420; So18G022460; So19G037390; So07G011940; So08G012630), _RT_ (So13G042380), _F3H_ (So22G037260), and _F3'H|F3'5'H_ (So01G039330). To analyze the change of lilac flower color, the sub-networks of differential expressed anthocyanins were performed (Fig. 5c; Supplementary Figs. 14–15). We found that Cy3Ru (Fig. 5c) and Cy3G (Supplementary Fig. 14) were two main high-content cyanidin-based pigments that caused red color fade, among the five analyzed anthocyanins distributed at T1 (Fig. 5c; Supplementary Figs. 14–15). The regulatory network of _F3'H|F3'5'H_ (So01G039330) was constructed (Fig. 5c), which encoded a key enzyme for the synthesis of DHQ and was directly related to the Cy3Ru. We initially speculated the potential regulatory networks in which WRKY played an important role together with MYB, bHLH, WD40 and ERF (Fig. 5c–d). Meanwhile, composition of the 5'UTR upstream cis-acting elements of the _F3'H|F3'5'H_ gene and the expression levels of related genes supported the networks (Supplementary Fig. 19).

Among the 425 potential direct regulatory factors at the transition stage (T2 and T3), the WD40 family contained most members (65), which was nearly two-thirds more than that at T1, followed by MYB and ERF families (Fig. 6a). Similarly, the potential regulatory network MYB, bHLH, and WD40 families and their targets (Supplementary Fig. 16), revealed a different situation from T1. At the transition stage, the structural genes with the most edges were _4CL_ (So16G047200), _C4H_ (So15G050380), _CHS_ (So16G045740), _CHI_ (So22G046560), _F3'H|F3'5'H_ (So07G050530), _DFR_ (So02G010260), _ANS_ (So03G007060), and _F3oGT_ (So09G055690; So01G036790) (Supplementary Fig. 16), most of which belonged to EBGs. Three anthocyanins including cyanidin (Cy), cyanidin-3-O-(2"-O-glucosyl)glucoside, and Pt3Ru were distributed at the transition stage (Fig. 6a). And, one structural gene (_4CL_, So08G002910) associated with Cy, and seven flavonols and anthocyanins enzyme genes (_PAL_, So03G064720; _4CL_, So13G051200; _CHS_, So19G016880; _FLS_, So13G055200; _F3H_, So15G003400; _F3'H|F3'5'H_, So05G012500; _F3oGT_, So18G014040) potentially regulated the Pt3Ru synthesis, which were with high expressions (PCC > 0.9 between metabolites and enzyme genes) (Fig. 6a; Supplementary Fig. 19).

Pt3Ru was the most important pigment between the transition and the fading stages (Fig. 4b). CHS was the first key enzyme in the synthesis and metabolism pathway of anthocyanins/flavonols, whose upstream regulation mechanism was unclear yet. Here, an important sub-network (Fig. 6b) inferred from TO-GCNs, showed that _CHS_ gene (So19G016880) with significantly high expression levels (Supplementary Fig. 19), was directly related to Pt3Ru, simultaneously, and was the potential first-order to third-order upstream regulators. In the co-expression network that we have resolved, _CHS_ gene (high expression levels in T2 and T3) might be regulated by _ERF_ So15G039730 (high expression level in T2) as an upstream third-level regulator, _GATA_ So01G024070 (high expression level in T2) as an intermediate secondary regulator, and _WD40_ So01G077230 (high expression level in T2 and T3) as the direct regulatory factor. The consistency of qRT-PCR data (Supplementary Fig. 19) and the result of DNA binding

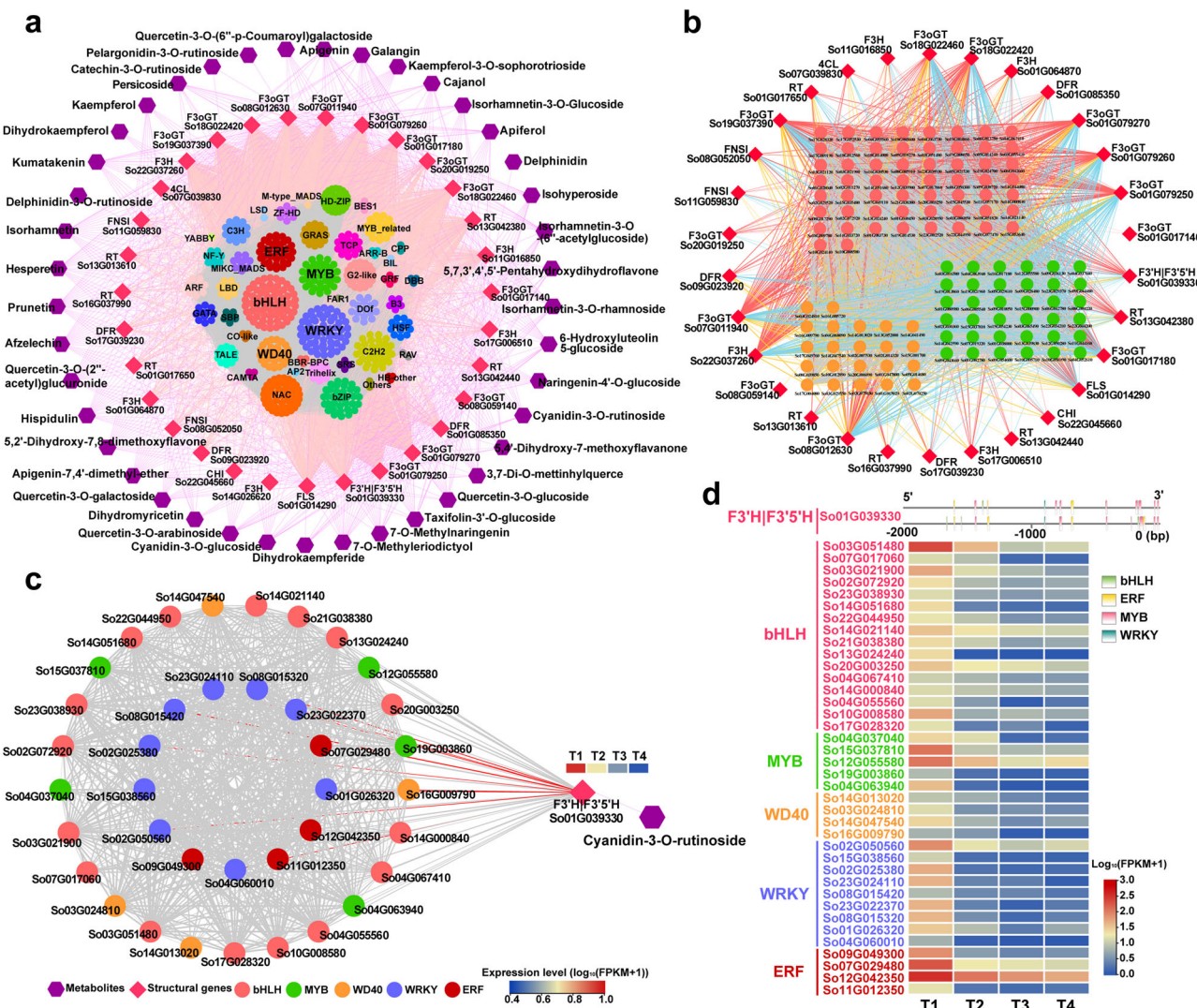

**Fig. 5 Sub-network for flavonoids metabolic pathways at the early stage of flowering. a** Sub-network for flavonoids metabolism. The network consists of metabolites (purple), structural genes (pink) and TFs from the outside to the inside. **b** Sub-network for flavonoids metabolism containing only the three families of TFs *MYB* (green), *bHLH* (pink), and *WD40* (orange) at the early flowering stage. The structural genes were linked to *bHLH*, *MYB* and *WD40* in red, blue, and orange lines, respectively. **c** Resolved one of the metabolisms of Cy3Ru and regulation for *F3'H|F3'5'H* gene involved. And 38 TFs (sixteen *bHLH*, five *MYB*, four *WD40*, nine *WRKY* and four *ERF*) identified as its direct regulators. **d** TFs (from Fig. 5c) heat-maps of gene expression profiles (log₁₀(FPKM+1)) at the four flower opening stages and DNA binding sites of *F3'H|F3'5'H* gene (from Fig. 5c) in 5'UTR upstream 2Kb. The bar color change from blue to red indicates the expression from low to high. The PCC value increases from small to large, the width of the edges changes from thinner to thicker in the networks.

site prediction may initially confirm our inference (Fig. 6c). At the same time, two *bHLH* family members, *So17G021180* and *So09G026360*, which exhibited significantly high and differential expressions during the four flower opening stages and were consistent with the expression level trend of *WD40 So01G077230* (Supplementary Fig. 19), maybe potentially interacted with the *WD40 So01G077230* accompanied by high PCC with 0.998 and 0.981. In addition, we also found that *ERFs* acted as both the direct regulators (*So11G003650* and *So18G001910*) and the upstream regulator (*So15G039730*), all with significantly high expressions (Supplementary Fig. 19). Furthermore, *F3'H|F3'5'H So05G012500* and *So07G050530*, which contributes to the red-based cyanidin and the blue-based delphinidin pigment, were analyzed for potential regulatory networks (Supplementary Figs. 17–18). Genes, *ERF So11G003650* and *ERF So15G039730*, were also identified as the important direct and upstream regulators in these networks.

## Discussion

In many horticultural plants, it is common for the flower color to fade or deepen with the flower opening. Although the research on anthocyanin accumulation has been very detailed, the metabolic dynamics and molecular regulation mechanisms of petal color change are still poorly understood. Not only does lilac flowers have the obvious color change, but the elegant purple color is an important ornamental trait and widely recognized. Here, we reported on a chromosome-scale genome assembly of *S. oblata* in the genus *Syringa* of family Oleaceae. Compared with the published genomes of Oleaceae plants (*F. excelsior*, *O. fragrans*, *O. europaea*, and *F. suspensa*), the present *S. oblata* genome assembly had the highest heterozygosity (1.92% vs 1.45% of *O. fragrans*, the highest known)[19], and the genome size (1.05 Gb) was less than *O. europaea* with 1.1[27]–1.48 Gb[20] and much larger than *O. fragrans* with 733.26[25]–740.71 Mb[19], *F. excelsior* with 866.8 Mb[18], and *F. suspensa* with 737.47Mb[32]. This assembled

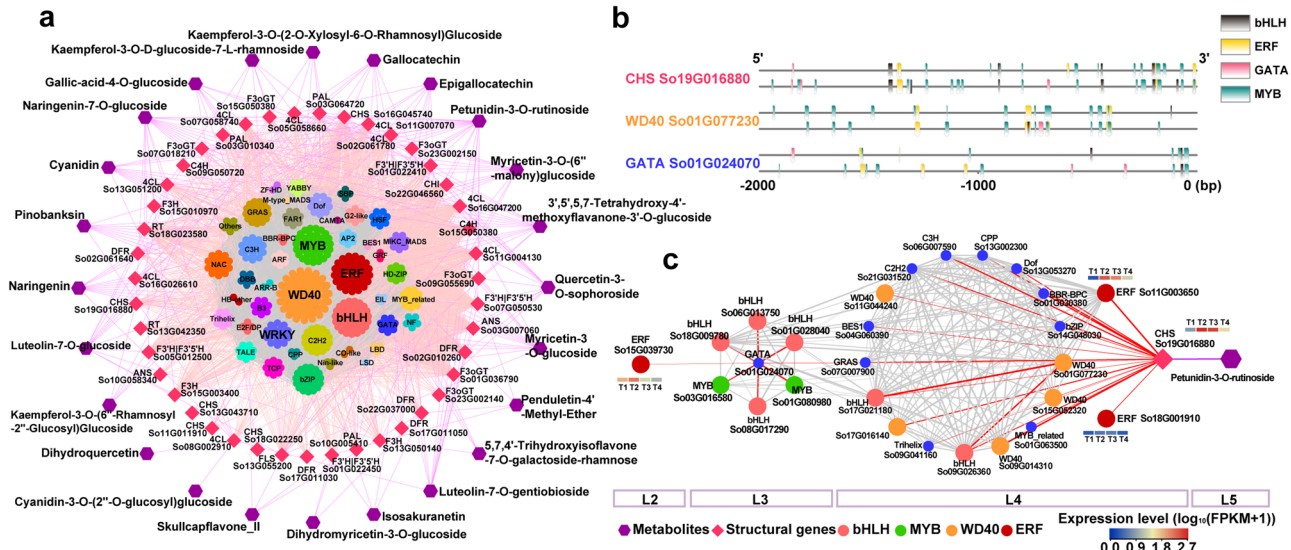

**Fig. 6 Sub-network for flavonoids metabolic pathways at the transition stage of flowering. a** Sub-network for flavonoids metabolism. The network consists of metabolites (purple), structural genes (pink) and TFs from the outside to the inside. **b** DNA binding sites of the one direct regulatory gene and two potential upstream regulatory genes in 5'UTR upstream 2Kb. **c** Resolved one of the metabolisms of Pt3Ru and hierarchical regulation for *CHS* gene involved. From right to left, the network consists of metabolite (Pt3Ru in L5), structural gene (*CHS* in L4), TFs that play a direct regulatory role (L4), and upstream regulatory TFs (L3 and L2). The expression profiles at the four flower opening stages were displayed as heat-maps (log₁₀(FPKM+1)). The bar color change from blue to red indicates the expression from low to high. The PCC value increases from small to large, the width of the edge changes from thinner to thicker in the networks.

genome is expected to be a valuable resource for genome-wide studies for genetic dissection of key traits, molecular breeding, and evolutionary studies in lilac.

Comparative analyses revealed that lilac experienced and shared two rounds of WGDs[20] along with *F. excelsior*, *O. fragrans*, and *O. europaea* within the Oleaceae family. The more recent WGD event is obviously shared with four Oleaceae plants, suggesting an Oleaceae-specific whole-genome duplication, which was consistent with previous reports for other Oleaceae species[18–20,27]. However, the older WGD event was barely visible may be due to the accelerated deletion of the oldest syntenic pairs after the recent WGD, a classically observed phenomenon[33] additionally supported by collinearity evidence. This lilac genomic information is expected to further clarify the evolution of Oleaceae plants[10].

Here, we performed the metabolome assay of time-ordered lilac flower samples to grasp the key flavonoid metabolites determining flower color change accurately and comprehensively. Metabolome results showed that Dp3Ru and Cy3Ru were the two major metabolites with the highest contents among the identified 118 flavonoid metabolites. Throughout the flower opening period, these two key anthocyanins showed distinctive and specific pattern of accumulation, distribution and fading. At the early stage of flower opening (T1), these two anthocyanins presented the highest relative contents, and the content of Cy3Ru was slightly higher than Dp3Ru (53.76% vs 46.24%); at the transition stage (T2-T3), the relative content of Cy3Ru was only two-thirds of T1, and on the contrary, the latter Dp3Ru was the main one (T2, 72.44% and 27.56%; T3, 70.25% and 29.75%); at the fading stage (T4), the lowest relative content was just one-sixth of T1, and Dp3Ru still was the main component with a proportion of 70.98%. Moreover, delphinidin-based anthocyanins were the major constituents of violet and blue flowers in nature, which has always been one of the most interesting topics in ornamental plant research, such as the world-famous cut flowers, roses, chrysanthemums, and lilies; such high contents of delphinidin-based anthocyanins

(Dp, Dp3Ru, Pt3Ru, Dp3Ru7G) in lilac petals provide ideas for future research.

It has been revealed that pelargonidin, cyanidin and delphinidin were attached to the B ring of the molecule with one, two and three hydroxyl groups, respectively, the increase of which would shift the color from redder to bluer[9]. Our analysis revealed that anthocyanin content was closely related to the lilac petals color, which shifted from reddish-purple to violet and then to light violet nearly white along with the flower opening period. This may be closely related to self-association or/and an intramolecular co-pigmentation[34]. In addition, glycosylation, including mono-, di-, and tri-saccharides, generally improved the stability and water solubility of the pigment[35]; with more stable diglycosides than their corresponding monoglycosides[36]; at the same time, location, number, and size of glycosidic substitutions played important roles on the anthocyanins color expression[37]. These also explained why the content of rutinoside (diglycoside) in the lilac petals was much higher than those of glucoside (monoglycoside) and pigments without sugar groups.

In particular, it is relatively rare to find anthocyanin rutinosides in ornamental plants, but more reports often focus on the fruits of plants with higher nutritional value, such as black currant[38], *Solanum betaceum*[39], and *Liriope platyphylla*[40]. Black currant is rich in anthocyanins and the content of anthocyanidin 3-O-rutinosides is significantly higher than that of anthocyanidin 3-O-glucosides; among which, Cy3Ru and Dp3Ru account for about 77% of the total anthocyanin content, and Cy3G and Dp3G account for about 21%. Anthocyanins extracted from black-currant fruit have been reported to be important in the treatment of cancer[41], cardiovascular disease and free radical scavenging, and are found intact in the blood when absorbed by humans as anthocyanin rutinosides[38]. In the petals of lilac, Cy3Ru and Dp3Ru account for about 98% of the total anthocyanin content, which may be of great value in future medical research and derivative products.

Based on the gene annotation and expression profile, we reconstructed the entire anthocyanin and flavonol metabolic pathways

and found there were multiple-copy genes for key enzymes playing major roles. The followed analyses of the time-ordered gene co-expression networks, combining transcriptomic and metabonomic data, unraveled important TF families and members which may play key roles in anthocyanins regulation in lilac petals. MBW-dependent (MYB-bHLH-WDR transcriptional complexes)[8,42,43] regulators were evidenced to control anthocyanin biosynthesis and meolism in our study, and WD40 seem to be more significant at the transition stage. It is worth noting that WRKY and ERF may play important roles at the early and transition periods, respectively. Several recent studies revetabaled the novel mechanisms by which WRKY TFs together with the MBW complex regulated the anthocyanin pigment pathway, and they even proposed the 'MBWW quartet' model[42]. Also, PyERF3 was found to interact with PyMYB114 and PybHLH3 to co-regulate fruits anthocyanin bio-synthesis in Asian red-skinned pears[43]. Therefore, we speculated that WRKY and ERF families, which were known to play crucial roles in plant tolerance against abiotic and biotic stresses, were potential regulators in flavonoid biosynthesis pathway[42,43], flower color change and patterning[12]. In addition, as the ethylene response factor, ERF was identified as the important direct as well as upstream regulators in our flower color change regulatory network, so we speculate that ethylene may affect petal color change, which may provide ideas for our future work.

In conclusion, the high-quality chromosome-level genome assembly presented here is expected to provide a reference for the further understanding of Oleaceae evolution and diversification. Furthermore, our transcriptome and metabolome investigations also provided an in-depth insight into lilac dynamic flower col-oration. Genomic data from this study will be an essential resource in many traits for further functional study with genome editing and also for molecular marker-assisted breeding to pro-mote genetic studies and novel cultivars breeding for *Syringa*.

## Methods

**Plant materials**. A 10-year-old *S. oblata* individual, with red-purple inflorescence, strong fragrance, and stable flower traits was chosen from the nursery of Beijing University of Agriculture (China) for genome sequencing and *de novo* genome assembly (Fig. 1). Fresh leaves were collected for the genome sequencing library preparation, and flowers at four blooming stages (Fig. 4a) were collected with three replicates to determine metabolites (flavonoids), and to perform transcriptome sequencing in support of genome annotation and gene expression analysis. Plant materials were immediately frozen in liquid nitrogen and stored at −80 °C for subsequent genomic DNA and RNA extractions.

**Library construction and sequencing**. Three kinds of libraries were constructed, including short-reads RNA-free sequencing with DNBSEQ™, long-read Nano-pore, and Hi-C, following manufacturers' instructions.

Petals' RNA was isolated using QIAGEN RNeasy plant mini kit (QIAGEN, Hilden, Germany) for short-read sequencing with DNBSEQ™. One library with insert sizes of 300–400 bp was constructed, and then a BGISEQ Platform was used to sequence the reads, resulting in 45.65Gb clean data, providing ~39.35× genome coverage.

High-quality and intact genomic DNA was extracted from fresh leaves using DNAsecure Plant Kit (Tiangen Biotech, Co. Ltd, Beijing, China). For long-reads sequencing, three Nanopore libraries with 20 kb insert size were generated and sequenced on the GridION X5 sequencer platform (Oxford Nanopore Technologies, UK) at the Beijing Genomics Institute (Wuhan, China). Hi-C library was sequenced on a BGISEQ platform.

**Genome size and heterozygosity estimation**. Size and heterozygosity of the lilac genome was estimated by *K*-mer analysis. Firstly, Jellyfish (v2.1.4) was used to quickly count the depth distribution of *K*-mer = 17–31, and the peak depth was clearly observed from the distribution data; then the frequency spectrum of *K*-mer was fitted by GenomeScope[44] (v1.0.0) to predict the genome characteristics.

**_De novo_ genome assembly and quality assessment**. *De novo* genome assembly employed the following steps: first, we used the NECAT (https://github.com/xiaochuanle/NECAT) to assemble the nanopore long-reads into contigs; subse-quently corrected three times using Racon[45] software; followed by mapping

short-reads to the reference genome using BWA-MEM[46] (v0.7.17), finally, fur-ther polished with the second round of pilon using clean DNBSEQ™ reads.

The Hi-C data were used for scaffolding the preliminary assemblies and to improve assembly continuity. The main Hi-C scaffolding workflow was as follows: clean Hi-C data was obtained by filtering Hi-C data from *Mbo*I digestion by Hic-pro software; then, clean Hi-C data were mapped to the corrected contigs genome by Juicer[47] (v1.5.6); 3D-DNA pipeline[48] was used for chromosome-level scaffolding by clustering, ordering, and orienting the previous assemblies based on genomic proximity information between Hi-C read pairs; finally, with the help of manual adjustment, the HIC cross-linking signal was redrawn by HiCPlotter[49] software and the adjacent anchored scaffolds were connected forming 23 superscaffolds corresponding to 23 pseudochromosomes.

Given that the *S. oblata* genome had higher heterozygosity, Purge_dup[50] was used to remove heterozygous and redundant sequences from the corrected genome to obtain corrected contigs. We evaluated the completeness of the assembled genome through BUSCO v5.2.2 analysis using gene content from the embryophyta_odb10[51], and mapping short-reads to the genome for quality assessment.

**Genome annotation**. First, repetitive elements were discovered by a combined strategy based on homologous sequence alignment (homolog) and *de novo* in the lilac genome. For homolog-based prediction, the RepBase[52] (v.21.12) library was applied to search against the lilac genome by RepeatMasker (v.4.0.7) (http://www.repeatmasker.org) and RepeatProteinMasker (v.4.0.7) with default parameters. For *de novo*-based approach prediction, a *de novo* repetitive element library was established with Piler[53], RepeatScout[54] and RepeatModeler, then predicted by RepeatMasker software. Furthermore, to identify the Tandem Repeats and LTR-RT, Tandem Repeats Finder (TRF)[55] (v4.09) and LTR_FINDER[56] (v1.06) were employed, respectively.

Gene structure prediction was based on a combination of evidence-based and *de novo* predictions. We used the Maker[57] software to carry out evidence-based assembly by evidence files, which included downloaded transcriptome data and mRNA/protein sequences of *O. europaea* subsp. *europaea*, *F. excelsior*, and *O. fragrans*. Transcriptome reads were sequenced on the BGISEQ-MGI2000RS platform and assemblies were generated (Supplementary Table 3) with Trinity (v.2.1.1) for genome annotation. *De novo* prediction was carried out using Augustus[58]. After repetitive sequence masking, the gene sets predicted by various methods were integrated into a non-redundant and more complete gene set. Finally, we used BUSCO (BUSCO, RRID:SCR_ 015008) to evaluate the quality of the gene set.

For non-coding RNAs (ncRNAs) annotation, the tRNAs were predicted using tRNAscan-SE[59] (v1.3.1) software. For highly conserved rRNAs, we chose related species' rRNA sequence as references, and predicted rRNAs sequences using BLASTN[60]. Through sequence information searches against the Rfam (https://rfam.org) database using INFERNAL[61] software, miRNAs and snRNAs were predicted.

With the help of eggNOG-mapper (http://eggnog-mapper.embl.de), gene functional annotation was performed using KEGG[23], GO[22] and COG[21] databases by comparing the eggNOG5.0 database. The advantages of eggNOG-mapper are: (i) ability to distinguish between paralogous and orthologous genes, and achieve more accurate gene function annotation and (ii) faster database searches through HMMER (http://www.hmmer.org) (v.3.0) or DIAMOND. In addition, functional annotation of protein-coding genes was carried out by performing BLASTP[60] (*E*-value ≤ 1e$^{-5}$) searches against SWISS-PROT[62], NCBI NR protein databases, MSU Rice Genome Annotation database[15] and Arabidopsis protein databases.

**Gene family, phylogenetic analysis, and divergence time estimation**. OrthoFinder[63] (v2.3.3) software was used to cluster paralogous and orthologous groups from genomes of other 11 species. CAFE (Computational Analysis of gene Family Evolution)[64] (v4.2.1) was used to analyze the expansion and contraction of gene family among these species. GO classification and KEGG pathways using the R package clusterProfiler[65]. After concatenating single-copy genes to form a super matrix following alignment by MUSCLE[66] (v3.8.31), the conservative regions from the generated matrix were extracted, and the gap information in the sequence was deleted to form a new matrix with Gblocks_0.91b[67]. Then RAxML (v8)[68] was used to construct the phylogenetic tree, and the FigTree program (v1.4.3) (http://tree.bio.ed.ac.uk/software/figtree/) was used for root determination. Codeml and MCMCtree programs, within the PAML package[69] (v4.5) software, were used to estimate the molecular clock (replacement rate) and the differentiation time between species. Finally, combined with the divergence time of known species from published literature or TimeTree database (http://www.timetree.org/) as the cali-bration (*S. lycopersicum-S. tuberosum* 5–9 Mya, *A. thaliana-C. papaya* 63–82 Mya, *O. europaea-O. fragrans* 7–45 Mya, *F. excelsior-O. fragrans* 33–47 Mya, *A. thaliana-O. sativa* 115–308 Mya), the lilac divergence time was calculated and estimated to be between 38–48 Mya.

**Genome collinearity/synteny and whole-genome duplication (WGD) ana-lyses**. For intergenomic comparison, we performed all-against-all BLASTP[60] with

$E$-value $\leq 1e^{-5}$ (v2.2.26) to align protein sequences, and the WGDI[70] and JCVI[71] (v0.8.12) package was used to determine high-confidence collinear blocks, paralogous and orthologous, in these species. Then, the $Ks$ (the number of synonymous substitutions per synonymous site) values of the homologs within syntenic blocks were calculated by PAML to identify WGDs. WGD time was estimated with *F. excelsior-S. oblata* divergence time (mean: 43.20 Mya) as an age constraint. $Ks$ peaks of *F. excelsior-S. oblata* syntenic orthologs allowed calculating $\gamma = Ks/(2 \times$ (divergence time))[33]. The same $\gamma$ value was applied to calculate the time of WGD events in lilac.

### Identification of flavonoid biosynthesis genes, transcription factors (TFs) and transcription factor binding site (TFBS)

Proteins in the flavonoid biosynthesis pathway including PAL (phenylalanine ammonia-lyase), C4H (cinnamate-4-hydroxylase), 4CL (4-coumarate CoA ligase), CHS (chalcone synthase), CHI (chalcone isomerase), F3H (flavanone 3-hydroxylase), F3'H (flavonoid 3'-hydroxylase), F3'5'H (flavonoid 3'5'-hydroxylase), FLS (flavonol synthase), DFR (dihydroflavonol 4-reductase), ANS (anthocyanidin synthase), F3oGT (flavonoide-3-O-glucosyltransferase), and RT (rhamnosyltransferase) were searched in the lilac genome annotation. Then, we added Hidden Markov Model (HMM) profiles of PAL (PF00221.20), C4H (PF00067.23), 4CL (PF00501.29, PF13193.7), CHS (PF02797.16, PF00195.20, PF00108.24), CHI (PF02431.16, PF16035.6), F3H (PF03171.21, PF14226.7), F3'H|F3'5'H (PF00067.23) (the F3'H|F3'5'H genes of other plants such as *V. vinifera*, *Petunia hybrida* and *A. thaliana* were also downloaded to construct phylogenetic tree to determine), FLS (PF03171.21, PF14226.7), DFR (PF01370.22, PF16363.6), ANS (PF03171.21, PF14226.7), F3oGT (PF00201.19), and RT (PF00201.19) from Pfam database (http://pfam.xfam.org/), to search against protein sequences databases employing the HMMER (v.3.0) in the *S. oblata* genome.

We used iTAK software[72] to identify TFs in the genome of the 12 assessed species. In addition, PlantRegMap was used to identify the WD40 family by homology to *A. thaliana*. The 2 kb sequences upstream of the genes were used to identify TFBSs present in the promoters of genes using PlantRegMap with threshold $p$-value $\leq 1e^{-4}$ (http://plantregmap.gao-lab.org/binding_site_prediction.php).

### Metabolome and transcriptome analyses

For the metabolome, after freeze-drying, crushing, weighting, dissolving, centrifugating, absorbing and filtrating, the samples were measured by Ultra Performance Liquid Chromatography (UPLC), and Tandem mass spectrometry (MS/MS).

Qualitative and quantitative analyses of the raw data obtained via UPLC-MS/MS were performed using Analyst v1.6.3 software. To determine the repeatability of metabolite extraction and detection, one quality control sample (QC) was inserted per ten samples, and then the overlapped total ion current chromatogram was analyzed. Principal Component Analysis (PCA), Fold Change (FC), and Orthogonal Partial Least Squares-Discriminant Analysis (OPLS-DA) were used to screen out differential metabolites (Supplementary Table 15), with FC > 2 or FC < 0.5, and variable importance in project (VIP based on OPLS-DA) greater 1. Then, we selected all eight anthocyanins and 12 additional flavonoids with high content in the differential metabolites for the PCA distance biplot.

Petals at four flower opening stages were collected with three replicates to perform transcriptome sequencing. For transcriptome analysis, clean reads were 304.3 million and Q20 and GC rate were 95% and 36.2%, respectively (Supplementary Table 3). DEGs (Supplementary Table 16) among the four flower blooming stages were identified using DESeq2, with adjusted $p$-value (FDR)<0.05 and $|\log_2 FC| \geq 1$.

### Regulation network construction based on TO-GCNs

We used a recently developed method of reconstructing TO-GCNs[73] for joint analysis of transcriptome and metabolome data. Finally, 80 differential metabolites and 20,984 DEGs (2507 TFs and 18,477 structural genes) were incorporated. The PCC cutoff values were 0.90 between structural genes and TFs, and 0.73 between metabolites and genes. TO-GCNs network was visualized using Cytoscape[74].

### Quantitative real-time PCR analysis

The key genes analyzed were verified by quantitative real-time PCR with the expression level of lilac *actin* gene as an internal reference. QRT-PCR was performed using Green qPCR SuperMix (TransStart) on a CFX Connect Detection System (Bio-Rad). Using the 2-$\Delta\Delta Ct$ method to calculate relative expression levels, and each analysis included three replicates. The specific primers for qRT-PCR were listed in Supplementary Table 17.

### Statistics and reproducibility

In Fig. 4b, Supplemental Figs. 10–12, and Supplemental Fig. 19, we used $n = 3$ biologically independent samples. Statistical analyses of metabolites relative contents and qRT-PCR were performed with GraphPad Prism 8. All the values were given as the mean ± SE and were analyzed by one-way ANOVA with multiple comparisons between adjacent stages (*$p <$ 0.05, **$p <$ 0.01, ***$p <$ 0.001, ****$p <$ 0.0001).

**Reporting summary**. Further information on research design is available in the Nature Research Reporting Summary linked to this article.

## Data availability

The genome assembly, genome sequencing, and RNA-Seq data used in the article have been deposited in the Short Read Archive under NCBI BioProject ID PRJNA766301, and FigShare[75]. Metabolome and data of Fig. 4b were provided in Supplementary Data 1 and 2.

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

## Acknowledgements

This study was supported by the National Key R&D Program of China (2018YFD1000406), the Joint Project of Beijing Municipal Education Commission and Beijing Municipal Natural Science Foundation (KZ201810020028). We thank Dr. Qiang Gao for assistance with the genome assembly.

## Author contributions

P.S.L., J.W., Z.H.H., and J.F.M. conceived and designed the study; B.M., T.L.S., J.W., Y.Y.Y., W.B.W., and W.B.X. prepared the materials, conducted the experiments, analyzed the data and prepared the results; B.M. wrote the manuscript; J.F.M., P.S.L., J.W., Z.H.H., Y.Z., S.C.S., Y.C.Y., I.P., and Y.A.E. edited and improved the manuscript; all authors approved the final manuscript.

## Competing interests

The authors declare no competing interests.
