## [Peer Review File · Communications Biology]

Reviewers' comments:

Reviewer #1 (Remarks to the Author):

In this manuscript, Ma and colleagues reported a genome of lilac and described the possible regulatory network of pigment biosynthesis in flowers. This work provide a valuable resource for research community and would be acceptable to publish in Communications Biology pending the authors' revision.

1. The authors should improve their writing to avoid obscurity of expression. For example, I don't think 'flowering' is appropriate in this work because it usually refers to flower transition process. Flower opening might be better.
2. Line 289-290, I don't understand the authors' interpretation here. Since Dp3Ru level is very low in T4 stage, how it be a key player in this stage? Please explain this.
3. Fig.4b, the histogram of Dp3Ru and Cy3Ru were presented in Supp Fig.10 and 11.
4. Line 332, I am wondering how the authors distinguish the 'indirectly regulated TFs'. Please explain it.
5. Line 347-349, Line 370-376, I don't think the data of Fig. 5 & 6 support these conclusions.

Reviewer #2 (Remarks to the Author):

1. Line 160~162: In the GO annotations, how many genes do assign to the biological process, cellular compound, and molecular function categories?
2. Line 176~177: "three sequenced woody flowering plants (*Prunus mume*, *Rhododendron delavayi*, and *Carica papaya*)", I don't think papaya is a woody plant. Papaya is a herbaceous plant with a soft-wooded single stem.
3. Line 626~627: How many repetitions do your RNA-seq experiment?
4. In Figure 4 (b), Figure S10, S11, and S12, I strongly recommend you need to do the real-time PCR analysis.

Reviewer #3 (Remarks to the Author):

Ma et al. reported the lilac genome assembly and used the newly assembled genome to provide insights into the molecular basis of petal color change underlying anthocyanin rutinosides. I have a few main comments that the authors could consider to improve the manuscript.

1. The heterozygosity of the assembly is quite high, 1.92%, which is much higher than most wild plant species. Maybe the assembled individual is a hybrid? If so, which may hugely influence the quality of the genome assembly. e.g. causing regional sequence expansion or collapse, which may be why the BUSCO (93.9%) is lower than most reported genome assembly currently. Given their dataset, I think the authors should put some effort to do a haplotype-level assembly and separate the assembly into two haplotypes using many relevant approaches, e.g. ALLHIC or many other available softwares
2. For the WGD analyses, the plot of Ks distribution in Figure 2C showed that some tandem gene pairs might contributing the small peaks at around 0 and the author could exclude genes with Ks values lower than 0.15. And I was wondering if the authors had seen a recent paper titled, WGDI: a user-friendly toolkit for evolutionary analyses of whole-genome duplications and ancestral karyotypes? Rather than JCVI, It may be more sensitive and accurate for the collinearity detection and dot plot of intra or inter genomic comparison. In addition, Figure 2d should show representative lines across these species, rather than only between *V. vinifera* and *S. oblata*
3. Structural genes which were highly regulated by TFs (lines 334 to 340) were considering as the key genes that are contributing the main high-content anthocyanins (in Figure 5c, Figure S14, Figure S15 and Figure S17). However, these genes might show a more lower expression values comparing to the other putative gene family members, for example, in Figure 3, the key gene So01G039330 presented more lower expression values than the other four homologous F3'H genes in almost four stages. Did the other homologous genes also show a compatible high correlation with TFs? Or the identification method of the flavonoid biosynthesis pathway should be

more accurate, because I was confused with the searching homologous genes method that the line 603 shows that PF00067.23 was employed to both F3'H and F3'5'H genes searching.

4. In addition, the authors used TO-GCNs to analysis the regulation network, why not use the gene-coexpression approaches like WGCNA, and/or some other temporal characteristics of transcriptome dataset, e.g. Mfuzz or k-means

Reviewers' comments and our responses:

#####

Reviewer #1:

In this manuscript, Ma and colleagues reported a genome of lilac and described the possible regulatory network of pigment biosynthesis in flowers. This work provide a valuable resource for research community and would be acceptable to publish in Communications Biology pending the authors' revision.

1. The authors should improve their writing to avoid obscurity of expression. For example, I don't think 'flowering' is appropriate in this work because it usually refers to flower transition process. Flower opening might be better.

Response: Yes, we improved the expression to change the word 'flowering' to 'flower opening' in revised manuscript.

2. Line 289-290, I don't understand the authors' interpretation here. Since Dp3Ru level is very low in T4 stage, how it be a key player in this stage? Please explain this.

Response: Thank you for pointing this out. We are very sorry, there is a potential misunderstanding in our presentation. What we intended to express here is that the differential expression of pigments suggests that cyanidin-based anthocyanins is the key pigment representing the color change from T1 to T3 and Dp3Ru may cause the color change from T3 to T4 in lilac petals. We have corrected the sentence in revised manuscript (see line 284-287).

3. Fig.4b, the histogram of Dp3Ru and Cy3Ru were presented in Supp Fig.10 and 11.

Response: We have removed the repeated histograms. Please see Supp Fig.10 and 11 in revised Supplementary Figures and Tables file.

4. Line 332, I am wondering how the authors distinguish the 'indirectly regulated TFs'. Please explain it.

Response: In this study, we are employing a time-ordered gene co-expression network (TO-GCN) (Chang et al., 2019) algorithm which could resolve the hierarchical gene regulation associated with a time-order developmental process. In the potential regulation networks predicated by this TO-GCN algorithm, we refer to TFs co-expressed with structural genes as the potential directly regulated TFs. Meanwhile, those clustered into the same network according to their expression levels, which have no direct co-expression relationship with structural genes but co-expressed with directly regulated TFs are identified as the indirectly regulated TFs. Here, we wanted to focus on TFs which co-expressed with the structural genes, and the expression in the original manuscript was removed to avoid overinterpretation (see line 330).

Chang, Y.M., Lin, H.H., Liu, W.Y. *et al.* Comparative transcriptomics method to infer gene coexpression networks and its applications to maize and rice leaf transcriptomes. *Proc Natl Acad Sci U S A*, 116(8), 3091-3099 (2019).

5. Line 347-349, Line 370-376, I don't think the data of Fig. 5 & 6 support these conclusions.

Response: Yes, we agree with the reviewer and have corrected it in line 344-347 and line 367-373. In addition, we provided the data of relative expression levels of related genes in T1-T4 estimated with qRT-PCR (Figure S19). We found the consistency of expression levels from different techniques confirmed the regulatory networks we resolved here.

#####

Reviewer #2:

1. Line 160~162: In the GO annotations, how many genes do assign to the biological process, cellular compound, and molecular function categories?

Response: According to statistics, there are 6,681 genes assigned to the biological process, 10,653 genes assigned to the cellular compound, and 6,958 genes assigned to the molecular function categories. And we have added this part of the description in revised manuscript, please see line 162-164.

2. Line 176~177: “three sequenced woody flowering plants (*Prunus mume*, *Rhododendron delavayi*, and *Carica papaya*)”, I don't think papaya is a woody plant. Papaya is a herbaceous plant with a soft-wooded single stem.

Response: Yes, thank you for pointing this out. We agree with the reviewer and have corrected it in line 174-175.

3. Line 626~627: How many repetitions do your RNA-seq experiment?

Response: Petals at four opening stages were collected with three replicates to perform transcriptome sequencing. And we have added it in revised manuscript, please see line 624-625.

4. In Figure 4 (b), Figure S10, S11, and S12, I strongly recommend you need to do the real-time PCR analysis.

Response: We have added qRT-PCR results for the key genes identified (including 5 structural genes and 7 TFs) as the reviewer suggested here, please see Figure S19.

#####

Reviewer #3:

Ma et al. reported the lilac genome assembly and used the newly assembled genome to provide insights into the molecular basis of petal color change underlying anthocyanin rutinosides. I have a few main comments that the authors could consider to improve the manuscript.

1. The heterozygosity of the assembly is quite high, 1.92%, which is much higher than most wild plant species. Maybe the assembled individual is a hybrid? If so, which may

hugely influence the quality of the genome assembly. e.g. causing regional sequence expansion or collapse, which may be why the BUSCO (93.9%) is lower than most reported genome assembly currently. Given their dataset, I think the authors should put some effort to do a haplotype-level assembly and separate the assembly into two haplotypes using many relevant approaches, e.g. ALLHiC or many other available softwares

Response: Thanks so much for this good question from reviewer.

1. We realized the same question when we performed the genome assembly. The genome size of the assembly prepared from ONT sequencing and NECAT is 1.12 G, which is very close to the predicted genome, so it seems the redundancy from mis-assembly of haplotypes is few, and haplotypes could not be resolved by using ALLHiC. And also ALLHiC requires a diploid contig backbone to the resolving the haplotypes, but we have no such data. Therefore, a haplotype-resolved assembly is not applicable and workable here, for this our study.

2. Now, we have a much higher BUSCOs score (96.60%) by running the BUSCOs evaluation. We have re-evaluated BUSCOs using the latest version of BUSCO 5.2.2 and the database OrthoDB v10 (Table S7 in revised Supplementary Figures and Tables). A total of 1614 genes (more than the previous 1292) were identified, and Complete BUSCOs was 96.60%. Duplicated BUSCOs of only 12.3% also suggests our genome assembly has high integrity with low redundancy and only a single set of genome was well-assembled.

3. Also a 99.37% mapping coverage rate of sequencing reads can also indicate the good integrity of the current assembly. We found a single set of haplotype genome of a high heterozygous species would have a lower BUSCO, just like the cultivated alfalfa, its four monoploid genomes contain 88.50, 88.30, 87.50, and 87.20% complete BUSCO genes, respectively (Chen *et al.*, 2020).

Chen, H., Zeng, Y., Yang, Y. *et al.* Allele-aware chromosome-level genome assembly and efficient transgene-free genome editing for the autotetraploid cultivated alfalfa. *Nat Commun* 11, 2494 (2020).

2. For the WGD analyses, the plot of Ks distribution in Figure 2C showed that some tandem gene pairs might contributing the small peaks at around 0 and the author could exclude genes with Ks values lower than 0.15. And I was wondering if the authors had seen a recent paper titled, WGDI: a user-friendly toolkit for evolutionary analyses of whole-genome duplications and ancestral karyotypes? Rather than JCVI, It may be more sensitive and accurate for the collinearity detection and dot plot of intra or inter genomic comparison. In addition, Figure 2d should show representative lines across these species, rather than only between *V. vinifera* and *S. oblata*

Response: Thanks for this very good comments. Here we provided the genome comparison (Ks analysis) by using WGDI and excluded genes with Ks values lower than 0.15 (in Figure 2c) in the revised manuscript. We also showed representative lines between *O. europaea* and *S. oblata*, *O. fragrans* and *S. oblata* (in Figure 2d).

3. Structural genes which were highly regulated by TFs (lines 334 to 340) were considering as the key genes that are contributing the main high-content anthocyanins (in Figure 5c, Figure S14, Figure S15 and Figure S17). However, these genes might show a more lower expression values comparing to the other putative gene family members, for example, in Figure 3, the key gene *So01G039330* presented more lower expression values than the other four homologous F3'H genes in almost four stages. Did the other homologous genes also show a compatible high correlation with TFs? Or the identification method of the flavonoid biosynthesis pathway should be more accurate, because I was confused with the searching homologous genes method that the line 603 shows that PF00067.23 was employed to both F3'H and F3'5'H genes searching.

Response:

1. Yes, thank you for pointing this out. We re-checked the NR annotation and found that *So02G039980* and *So12G044550* genes were annotated to flavonoid 3',5'-methyltransferase [*Olea europaea* var. *sylvestris*], so these two genes were deleted in the revised manuscript (in Figure 3); both F3'H and F3'5'H genes belong to the CYP75 subfamily in the P450 gene family, so, in addition to using PF00067.23 (P450 gene family) for searching homologous genes, we also downloaded the CYP75 genes of other plants such as *V. vinifera*, *Petunia hybrida* and *Arabidopsis thaliana* to construct a phylogenetic tree, which confirming that *So01G039330*, *So05G012500*, *So07G050530*, *So01G022410* and *So01G022450* belong to the F3'H|F3'5'H gene family (CYP75 subfamily); F3'H and F3'5'H genes are very close in homology and are difficult to separate, so we collectively refer to them as F3'H|F3'5'H genes in the revised manuscript, the same is true for the azaleas (Yang, *et al.*, 2020).

2. In our analysis, the networks were divided into eight time-ordered expression levels (L1-L8) by the expression trends of metabolites, structural genes and transcription factors, and we wanted to focus on the **anthocyanin** metabolites sub-network in T1 (L1-L3, Figure 5) and T2-T3 (L4-L6, Figure 6). For the five F3'H|F3'5'H genes, *So01G039330* belongs to T1 (L1-L3, we have shown in Figure 5), and *So05G012500* (we have shown in Figure S17) and *So07G050530* (we have supplemented in Figure S18 in the revised manuscript) belong to T2-T3 (L4-L6); while *So01G022410* and *So01G022450* were highly expressed in T4, and the PCC (Pearson's correlation coefficient) between them and the anthocyanins metabolites we were interested in was not within the threshold, therefore their sub-networks were not performed.

3. In addition, we validated the expression levels of *So01G039330*, *So05G012500* and *So07G050530* genes at different flower opening stages (T1-T4) by RT-qPCR, the results were consistent with the transcriptome, which supported that we divided time-ordered expression levels, please see Figure S19.

Yang, F.S., Nie, S., Liu, H., *et al.* Chromosome-level genome assembly of a parent species of widely cultivated azaleas. *Nat Commun*, 11(1), 5269 (2020).

4. In addition, the authors used TO-GCNs to analysis the regulation network, why not

use the gene-coexpression approaches like WGCNA, and/or some other temporal characteristics of transcriptome dataset, e.g. Mfuzz or k-means

Response: In our study, the changes of anthocyanin content in lilac were closely related to the changes of petal color during flower opening, so the joint analysis of metabolome and transcriptome data can better solve the problem of related molecular regulatory mechanisms. Although both WGCNA and TO-GCNs meet this requirement, considering that flower color change during flower opening is a continuous and dynamic process, time-ordered sampling gene co-expression analyses (TO-GCNs) (Chang et al., 2019) are more suitable than WGCNA, so that, TO-GCNs was selected.

Chang, Y.M., Lin, H.H., Liu, W.Y. *et al.* Comparative transcriptomics method to infer gene coexpression networks and its applications to maize and rice leaf transcriptomes. *Proc Natl Acad Sci U S A*, 116(8), 3091-3099 (2019).

REVIEWERS' COMMENTS:

Reviewer #1 (Remarks to the Author):

The authors have addressed all my concerns. I think this version is ready for publication.

Reviewer #2 (Remarks to the Author):

I agree with the author's reply.

Reviewer #3 (Remarks to the Author):

I thank the author for putting effort to resolve all my concerns. I have no more questions.